# Potential Effects of Environmental and Occupational Exposure to Microplastics: An Overview of Air Contamination

**DOI:** 10.3390/toxics12050320

**Published:** 2024-04-28

**Authors:** Priscilla Boccia, Simona Mondellini, Simona Mauro, Miriam Zanellato, Marco Parolini, Elena Sturchio

**Affiliations:** 1INAIL—Istituto Nazionale per L’Assicurazione Contro gli Infortuni sul Lavoro, Dit, 38/40 Via Roberto Ferruzzi, 00143 Rome, Italy; m.zanellato@inail.it (M.Z.); e.sturchio@inail.it (E.S.); 2Department of Environmental Science and Policy, University of Milan, Via Celoria 26, 20133 Milan, Italy; simona.mondellini@unimi.it (S.M.); marco.parolini@unimi.it (M.P.); 3Chemistry Department, University of Rome “La Sapienza”, Piazzale Aldo Moro 5, 00185 Rome, Italy; mauro.1856617@studenti.uniroma1.it

**Keywords:** microplastics, legislation, exposure, atmospheric contamination, toxicity, human health

## Abstract

Microplastics (MPs) are now ubiquitous environmental contaminants that lead to unavoidable human exposure; they have received increasing attention in recent years and have become an emerging area of research. The greatest concern is the negative impacts of MPs on marine, fresh-water, and terrestrial ecosystems, as well as human health, to the extent that the World Health Organization (WHO) calls for increased research and standardized methods to assess exposure to MPs. Many countries and international organizations are implementing or proposing legislation in this regard. This review aims to summarize the current state of legislation, indoor and outdoor contamination, and potential human health risk due to exposure to airborne MPs, considering that occupational exposure to MPs is also becoming a growing area of concern. Even though research regarding MPs has continuously increased in the last twenty years, the effects of MPs on human health have been scarcely investigated, and toxicity studies are still limited and not directly comparable, due to the lack of standardized studies in this field.

## 1. Introduction

Microplastics (MPs) are ubiquitous contaminant particles which have a diameter lower than 5 mm in size, and are found in various shapes, including microbeads, microfibers, and fragments. A more recent definition sets the lower and upper limits of the MP size range between 1 μm and 1 mm, respectively [1]. According to the 2018 European Regulation, MPs can be divided into two categories based on their source or origin, specifically primary and secondary MPs. Primary MPs consist of plastic items released directly into the environment in a micrometric size. The main source of this typology is the washing of synthetic garments (35%), tire abrasion during driving (28%) and MPs intentionally added to cosmetic products (2%) (the European Parliament). Primary MPs represent 15–31% of all those present in the ocean. Secondary MPs are plastic items deriving from the progressive decomposition of large-size plastic materials or waste. They represent approximately 68–81% of MPs present in the ocean [2]. The term “microplastics” was coined for the first time 20 years ago by Thompson et al. [3], who investigated ocean pollution caused by plastics in the UK, resulting in numerous publications on this topic. Researchers have been worried about the potential risk of MPs for ecosystem health, and they have accumulated extensive and deeply concerning evidence of MPs’ negative impacts on marine, freshwater, and terrestrial fauna, flora, ecosystems, and habitats, as well as recently on human health [4,5,6,7]. Most of the literature is related to the presence of MPs in marine waters, of which 80% derive from anthropogenic land activities. However, their presence has also been detected in the air and in the soil, with effects on the chemical–physical properties of these matrices [8]. Although several studies have pointed out the adverse effects induced by exposure to MPs on diverse aquatic and terrestrial organisms [9], to date, little is known about their health effects on humans [10]. Thus, in recent years, MPs have received increasing attention, becoming an emerging research area. In fact, in 2019, the World Health Organization (WHO) demanded that research be strengthened in this field and highlighted the need to develop standardized methods to obtain a more accurate assessment of exposure to MPs. MPs can cause remarkable ecological and human health-related concerns due to their environmental persistence, potential ecotoxicity, and their capability to act as carriers of chemical pollutants and pathogens. Regulation of plastic and MP pollution has become a significant focus worldwide, as awareness of their environmental impact has increased. MPs are known for their persistence in the environment, their potential to harm wildlife, and for entering the food chain, creating risks to human health. For these reasons, numerous countries and international agencies have implemented or proposed regulations to mitigate the production, use, and disposal of plastics and MPs.

## 2. Regulation on Microplastic Pollution

At the global level, the United Nations Environment Programme (UNEP) has been dealing with global action against plastic pollution, including MPs, through the development of a global treaty aimed at disposing plastic waste, with ongoing discussions and negotiations among member states. Furthermore, solid plastic waste was introduced in the Basel Convention (which aimed to limit global trade in hazardous waste), regulating the international trade in plastic waste [11]. On 27 July 2017, China issued a ban on importing 24 types of solid waste, including plastic waste [12,13].

The United States of America (USA) is taking action with international agencies to address plastic pollution, aiming to reduce plastic use in several sectors and its presence globally, and to enhance global engagement and improve domestic infrastructure for recycling and reducing litter. In the USA, the distribution of cosmetics containing plastic microspheres [14] has been banned since 2019. California has banned single-use plastic bags and implemented a law requiring all packaging to be recyclable or compostable by 2032 [15].

The European Union (EU) has been a leader in regulating plastics, adopting several actions under the European Green Deal and the Circular Economy Action Plan. New strategies took place for plastics in the circular economy, aiming to make all plastic packaging reusable or recyclable by 2030, such as the notable Single-Use Plastics Directive [16], which bans certain single-use plastic items (like cutlery, plates, straws, and cotton bud sticks) and implements measures to reduce the use of others. Many of the countries belonging to the EU are considering restrictions on MPs under the program for the Registration, Evaluation, Authorization, and Restriction of Chemicals (REACH). The action focuses primarily on restrictions in products, and some countries have implemented initiatives at a local level (Figure 1).

Worldwide, specific legislation on MP pollution has evolved, because new regulatory measures or mitigation strategies are being considered based on different approaches, such as on microbead restrictions in products, packaging, and promoting biodegradable alternatives.

In 2017, the European Commission invited the ECHA (European Chemicals Agency) to evaluate the scientific evidence for taking regulatory action at the EU level on intentionally added MPs. In January 2019, the ECHA proposed the broad application restriction of MPs in products placed on the European market to avoid or reduce their release into the environment [17]. The European Commission is also evaluating other options to reduce the release of MPs accidentally formed in the aquatic environment to create a new action plan in the circular economy. In 2018, the EU Strategy for Plastics in a Circular Economy acknowledged the risks posed by MPs and advocated innovative solutions. In 2020, as a follow-up action of the European Green Deal [18], the Commission faced the presence of MPs in the environment by restricting intentionally added MP in products and addressing unintentional releases of MPs by developing standardization and regulatory measures, as well as harmonizing methods for measuring their releases. This is because one of the targets of the Green Deal’s Zero Pollution Ambition is to significantly reduce plastic litter and MPs [19], and to provide further data on MP concentrations ain other environmental compartments, such as surface waters. On 16 October 2023, the Commission put forward a proposal for a regulation to reduce MP pollution from plastic pellet losses. Plastic pellets (also called nurdles, nibs, preproduction pellets, and resin pellets) are the industrial raw material used for all plastic production. In Italy, Article 9 of bill 2582 provided for a ban on the trade of cosmetic rinse-off products with exfoliating or cleansing action containing MPs from 1 January 2020 [20]. In 2021, in its action plan named ‘Towards zero pollution for Air, Water and Soil’ [19], the Commission proposed that, by 2030, the EU should reduce (intentional and unintentional) MP releases into the environment by 30%.

On 25 September 2023, a provision was issued by the European Commission, which is included in REACH, i.e., the European regulation that deals with the registration, authorization, and restriction of chemical substances in the European Union (Regulation (EC) No 1907/2006, [21]). More precisely, an amendment was made to Annex XVII of Regulation (EC) No. 1907/2006 of the European Parliament and Council: it prohibits the marketing of most MPs, including glitter, diamonds, and microspheres [22] (Figure 2).

A critical and still unexplored issue, from an occupational point of view, concerns the potential risk and the related mode of exposure of workers to MPs. In the workplace, the exposure could be of many orders of magnitude in terms of concentrations compared to that of the general population, which could be exposed to low concentrations of MPs in the air; as such, it represents a major exposure route [23,24]. The most exposed workers belong to the waste management industries, waste recycling operations, plastic and composite production, and the production of polyvinyl chloride (PVC) pipes [25]. Additional categories are workers involved in the smoothing and processing of plastic and in the textile industry, i.e., the workers who cut polymer fibers, such as nylon flocking cutting [26].

The National Institute for Occupational Safety and Health (NIOSH) identified different pathways of MPs that can interact with humans through inhalation in the workplace, including mechanical and environmental degradation of plastics during waste management and recycling operations, degradation of carpets and other synthetic products (releasing fibers), shredding of polymers and plastic products (generating dusts), high-energy or high-heat (e.g., laser cutting or high-speed drilling) treatment of polymer composites, 3D printing from the melting or fusing of plastics, and industries hosting plastic processers and printers [27]. NIOSH recommends mitigating exposure for their employees through appropriate controls in their workplaces, since there are no existing regulations for nano- and micro-plastic workplace contaminants.

Given the lack of relevant scientific data on the effects of MPs on human health and the lack of standardized methods for collection, isolation, separation, identification, and quantification of MPs, additives, and chemical substances within complex mixtures, it is important to develop new protocols for the analysis and characterization of the most commonly used MPs (textiles, cosmetics). Therefore, it is important to collect further data and to develop and standardize new methods for evaluating the levels of MP and/or additive chemical substances present in different environmental compartments or matrices, including indoor environments and workplaces.

## 3. Detection Limits in Different Matrices

Specific emission detection limits for MPs in different matrices, such as air, water, soil, or sediment, have not been universally established and standardized. However, research and discussions about MP pollution are ongoing, and guidelines or regulations may evolve.

Different countries and regions might have their own monitoring programs and research initiatives to assess the levels of MP in different environmental matrices. Researchers often use a variety of sampling and analytical techniques to quantify and characterize MPs in different media. These methods may involve visual identification, spectroscopy, or other advanced laboratory techniques.

In the Directive (EU) 2020/2184 “on the quality of water intended for human consumption”, the term MP appeared for the first time [28]. Described as emerging compounds, MPs have been related to the “watch list” mechanism introduced with Directive (EU) 2020/2184. The European Commission, in 2019, submitted a report of risk analysis related to MPs in drinking water and, by 2024, aimed to adopt an analytical method to measure MPs.

European Directive strategies pointed out the impacts of MPs on marine ecosystems, such as disturbing their ‘good environmental status’. Plastics of different sizes, shapes, and polymer compositions can affect and damage ecosystems and enter into the human food chain, posing risks to public health. Additional scientific research and the implementation of policy measures are necessary to address the severe threats caused by MP pollution to global ecosystems and human health. Mitigating plastic impacts on the health of people, animals, and ecosystems requires an approach that recognizes how people, animals, and plants are interconnected and how their individual health is itself dependent on the health of their shared environment [29] (Figure 3).

Moreover, regulatory agencies are increasingly considering air as an important route for MPs. The United Nation Environmental Protection Agency (EPA) has promulgated an action for national emission standards for hazardous air pollutants (NESHAP) for new and existing reinforced plastic composite production industries. In a recent report on MPs in the environment, the German Environment Agency [30] identified the major source of MPs in air as being from tire wear. In a draft of a science report on plastic pollution by Environment and Climate Change Canada and Health Canada [31], the presence of MPs was studied in outdoor and indoor air compartments, and the main MP sources were identified as fibers from textiles and wear particles from tires.

The plastic and MP regulatory landscape is rapidly evolving, with new policies and innovations continually emerging, contemporary with the development of knowledge of their important implications for environmental health, biodiversity, and sustainable development.

**Figure 3 toxics-12-00320-f003:**
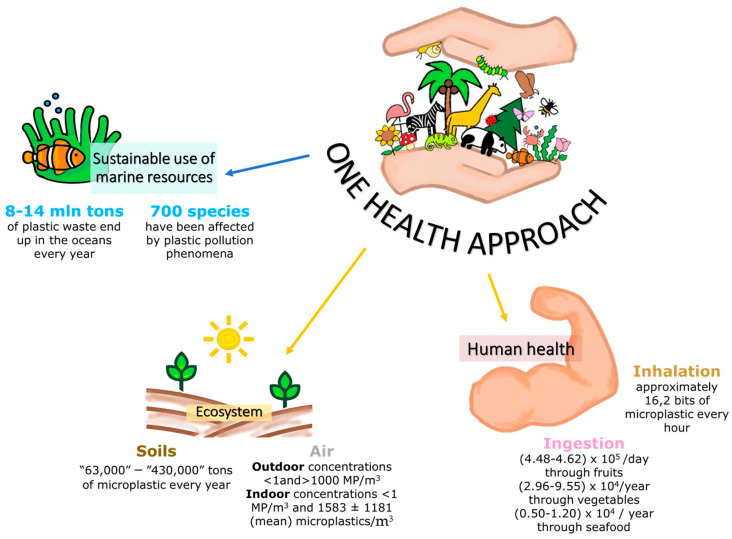
Plastic impacts on the health of people, animals, and ecosystems, demonstrating how these are all interconnected. Data from [32,33].

## 4. Analytical Methods

There are various methods and techniques used to qualitatively and quantitatively determine MPs in environmental matrices (Table 1). In most studies, MPs are first identified visually through a stereomicroscope [34]. This approach does not allow the identification of the chemical composition of the isolated items with certainty, but it can be considered a screening selection step of putative MPs. One of the problems with this type of analysis is the possibility of mistakenly confusing sand or carbon particles as MPs or vice versa. A possible solution to improve MP detection under the microscope is the use of an apolar dye capable of binding to MPs and not to the inorganic component of the sample. The most widely used dye for these purposes is the Nile Red, which after binding to plastic polymers emits fluorescence under a fluorescence microscope [35]. An alternative dye is Rose Bengal; in contrast to Nile Red, being a hydrophilic dye, it binds to inorganic particles and not polymeric ones, which do not emit fluorescence under fluorescence microscope [36]. Light microscopy allows for detecting putative MPs by observing their shape and size but, in the case of items smaller than 10 μm, it is very difficult to discriminate whether they have polymeric or mineral composition. The solution is to rely on scanning electron microscopy (SEM). This instrument returns much more detailed information about the morphological characteristics of the sample under examination. When combined with X-ray spectroscopy (SEM-EDS), it is possible to obtain higher resolution images, to determine the elemental composition of the sample, and to discriminate items of inorganic origin from carbonaceous ones [37]. However, visual identification is now considered outdated because it often provides insufficient results and false positives. Thus, it is preferred to use spectroscopic or spectrometric techniques to identify MPs. One method consists of using pyrolysis combined with a gas chromatograph coupled with mass spectrometer (Pyr-GC/MS), which analyzes vapors developed through the pyrolysis of polymers and the evaporation of additives and organic contaminants in the samples. It is a destructive method that prevents any further analysis of the sample, and it does not allow researchers to determine other parameters, such as shape, size, and number of items. However, it has the advantage of simultaneously analyzing all MPs by quantifying their total mass [38] (Table 1).

Among the spectroscopic methods, there are vibrational infrared and Raman spectroscopies. Infrared spectroscopy is based on the absorption of electromagnetic radiation in the infrared zone, which determines molecular vibrations and rotations. This technique allows the immediate identification of functional groups present in a molecule, just by observing the spectrum. This feature, together with a simple data recording technique, makes IR spectroscopy the easiest, fastest, and often most reliable method of assigning a substance to a particular class of compounds.

Raman spectroscopy is commonly used to determine vibrational modes of molecules and to provide a structural fingerprint by which molecules can be identified. The Raman technique has the advantage that water does not give any interference, and the signals are generally narrower and, therefore, easier to identify.

These two spectroscopic techniques are considered suitable for the analysis of MPs and can also be combined with a microscope for a more specific analysis [39,40]. Both spectroscopies return a spectrum from which it is possible to understand the chemical composition of the particle by comparing it with reference spectra of pure plastic materials in specific databases. These two techniques are defined as complementary, since Raman is more sensitive to the vibrations of homopolar bonds, such as C-C, while IR is more sensitive to those of functional groups with polarized bonds, e.g., C=O and others. In fact, in a centrosymmetric molecule, all the vibrations that are symmetrical with respect to the center of symmetry are inactive in the IR spectrum (prohibited by the selection rules), since they do not generate variations in the dipole moment. However, these oscillations are active during Raman spectroscopy, as they change the polarizability of the molecule.

However, the identification of polymers by Fourier transform infrared (FTIR) and Raman spectroscopy is, for example, susceptible to the presence of additives on the polymer surface. Dust, microbial fouling, and dyes on the plastic material can interfere with the signals and lead to errors in substance identification.

## 5. Microplastic Exposure

The ubiquitous presence of MPs in different environmental compartments (water, soil, and air) results in human exposure through several routes, such as diet, drinking water, inhalation, and skin contact (Figure 4).

### 5.1. Exposure through the Diet

Several studies have identified the presence of MPs in table salt extracted from oceans, lakes, and salt rocks in different countries worldwide [41]. Salt is an essential food within the diet: the World Health Organization (WHO) estimates that an adult consumes approximately 9–12 g of salt each day. During the process of salt crystallization, water evaporates, while MPs are retained within the crystals. Based on data obtained from 11 papers, an adult can ingest about (5.00–7.00) × 10^3^ MPs in a year through salt consumption [41]. MPs in seafood products are also an important pathway of exposure. The marine environment represents the main sink for plastic waste, with approximately 80% of marine plastic debris on average entering the oceans via riverine pathways due to human activities. Strokal et al. [42] estimated that rivers export approximately 0.5 million tons of plastics per year worldwide. In the marine environment, MPs are ingested by aquatic organisms, which can experience bioaccumulation [43]. Then, MPs can move over the trophic levels, starting from zooplankton, passing through smaller fish, then larger fish, and finally reaching humans who feed on them. MPs have been found in diverse bivalves, such as mussels, clams, and oysters, but also in fish and marine mammals. Statistical analyses have shown that, on average, each person consumes about 2.4–4.8 kg of shellfish and 7.3–13.7 kg of seafood products in total in a year [41]. Based on data obtained from 18 representative articles, it is estimated that an adult can ingest approximately (0.50–1.20) × 10^4^ MPs in a year through seafood products [41].

Another source of MP could be crops; it has been estimated that at least 473,000 tons of plastic waste is released into the soil each year in European Union countries [44]. The presence of MPs in soils is due to different sources, such as mulching and sewage irrigation in agricultural fields. The amount may vary from approximately 44,000–300,000 to 63,000–430,000 tons of MPs through sewage sludge applied annually to North American and European farmland, respectively [45]. MPs in soil can be largely adsorbed by plants through their roots; once adsorbed, they can migrate and reach as far as the stems and leaves that can be consumed. MP accumulation was observed in lettuce, wheat, and rice [46]. A recent study compared the number of MPs in fruits and vegetables, such as carrots, broccoli, potatoes, apples, and pears, and determined that apples and carrots are the most contaminated species [47]. The same article reported that the daily number of MPs ingested through fruits is approximately (4.48–4.62) × 10^5^, while through vegetables it is (2.96–9.55) × 10^4^ MPs a year for adults.

The release of MPs from plastic food containers and bags under different conditions was analyzed [48]. The results highlight how heating inside the microwave oven leads to the release of a greater number of MPs inside food in comparison with storage in the freezer or refrigerator. It was observed that some containers could release 4.22 million MPs from an area of 1 cm^2^ in just 3 minutes of microwave heating. Storage at both low temperature and room temperature can also release millions to trillions of MPs. In addition, food bags made of polyethylene have also been found to release more particles than those made of polypropylene [48].

### 5.2. Ingestion through Drinking Water

The contamination of drinking water by MPs could be the most dangerous compared to all other routes of exposure, given the greater amount of water introduced into the body daily.

MPs were initially found in tap water [49], and many subsequent studies have also confirmed their presence in bottled water and beverages, beer, and tea [50,51,52]. A higher level of MP contamination has been found in bottled beverages compared to tap water due to the industrial production and packaging processes undergone by the latter. In addition, water contained in 22 recyclable and single-use plastic bottles was compared with water from 9 glass bottles and 3 cartons [53]. It was observed that MPs in glass bottles were in lower amounts than those in plastic bottles, and that single-use plastic bottles and cartons both contain lower numbers of MPs than recyclable plastic bottles. This is because repeated use of the same bottle promotes its wear and tear and, consequently, the release of more MPs.

### 5.3. Skin Contact

It is believed that MPs do not pass through the skin barrier [54], but the exposure risk is increased by their prolonged deposition. MPs have been detected in atmospheric fallout, resulting in deposition on human skin and dermal exposure [55]. For example, the use of consumer products containing MPs (such as face creams and facial scrubs) will increase the exposure risk [56]. Protective mobile phone cases can generate MPs during use, which are transferred to human hands [51].

### 5.4. Exposure through Inhalation

In the last twenty years, research regarding MPs has continuously increased. Initially focusing mainly on the marine environment and its biota, researchers are now investigating the prevalence and effects of MPs in other environmental compartments, including the atmosphere [57]. Because MPs have a small size, their low material density and high surface area can determine their air suspension and potential dispersion via atmospheric agents [58]. The first studies reporting MPs in the atmosphere were published in 2015 [59,60]. Since then, the number of studies on the topic have increased. The main sampling methods employed in these studies are wet and dry deposition, atmospheric sampling, and dust collection. This difference in methodology, together with the different concentration metrics that are used, hampers study comparability [61]. Most of the studies confirmed fibers to be the most abundant MP type in both indoor and outdoor environments [62,63], together with tire wear particles (TWP) and road dust in the latter environment [58].

#### 5.4.1. Microplastics in Outdoor Environments

In outdoor environments, MP concentration varies with geographic distribution and land use, and it is related to the presence of urban areas and high population densities. Generally, outdoor air MP concentration results are lower than those from indoor environments [59,64], probably due to a greater use of textiles (e.g., clothing, carpets, fabric furniture) and lower ventilation in indoor environments. Airborne MPs appear to fluctuate in combination with meteorological factors, such as precipitation events and winds [61], which also play an important role in the global transport of MPs to terrestrial and aquatic environments [65]. Different studies, in fact, have reported that air deposition of MPs is higher during wet periods and decreases during those of dry weather [66,67] because of wet scavenging. Snow or water droplets can trap particles and so remove them from the atmosphere [59,61,68]. In contrast, other studies have suggested that wind action and not exclusively wet deposition might be the main agent determining atmospheric MP removal via dry deposition and dispersion [67,69] (Table 2).

Important sources of outdoor MPs are textiles, tire abrasion, urban dust, paint, construction sites, the incineration of urban waste, wastewater sludge for agricultural use and mulching films, and open dumps and landfills [6,65,70]. Concentrations reported in the scientific literature vary with the sampling region and time of the year. The first investigation by Gasperi and co-authors [60] employed an active sampling method to filter the air of outdoor and indoor environments within the urban area of Paris (France). Their study highlighted that the main component of airborne MP is constituted of fibers in the sub-millimetric size range (80% between 100 and 500 µm) and that their concentration is higher in indoor environments (3–15 particle m^−3^) than outdoors (0.2–0.8 particle m^−3^). In a study conducted in Southern Iran [62], 16 suspended dust samples were collected during the dry season from urban and industrial areas. The sample collection was carried out over eight consecutive days using polytetrafluoroethylene (PTFE) filters (2 µm pore size), and the samples were analyzed via optical and scanning electron microscopy (SEM). The number of collected MPs ranged from 0.3 to 1.1 particles m^−3^ and was equally distributed between industrial and urban areas; of these nearly all, excluding six items, were fibers (n = 214). Another work was conducted by the same group in Ahvaz (Iran) [71]. In this study, sampling was performed at two sites for sixteen days, distributed over four months, using glass fiber filters (1.6 µm pore size) and an active air sampler. PM_10_ (particulate matter with a diameter of 10 mm or less) was selected as the size cut off-for the analyses. Collected particles were visually inspected after a density separation step (ZnCl_2_). The MP concentrations were similar in both locations (0.002–0.007 particle m^−3^, 23–341 particle g^−1^ of PM_10_ in site 1; 0.002–0.015 particle m^−3^, 34–162 particle g^−1^ of PM_10_ in site 2). Most of the investigated particles were fibers of size 15–35 µm; Raman investigations on 19 particles showed that polyethylene terephthalate (PET), polypropylene (PP), nylon, and polystyrene (PS) were found to be the most represented polymers. Precipitation, wind speed, and time did not seem to have an impact over the MP concentration during this study. Higher MP concentrations have been reported in the city of Shanghai (China) [63], where a study reported values from 0 to 4.82 particles m^−3^ (on average 1.42 ± 1.42 particles m^−3^) and estimated an annual weight of suspended MPs of approximately 120.72 kg. Fibrous MPs represented 67% of the sample, followed by 30% fragments and 3% granules. In this study, particles were collected in triplicate with sampling stations, six of which were placed in different municipal districts of the city area, whiles three more were placed at different heights (1.7, 33, and 80 m) on a building in the East China Normal University. Suspected MPs were first visually inspected, then analyzed via micro-FTIR (µ-FTIR). Spectroscopic analyses showed that 49% of the collected MP were comprised of PET, polyethylene (PE), and polyether sulfone (PES), followed by polyacrylonitrile (PAN) (12%) and polyacrylic acid (PAA) (9%). Another recent study conducted in Beijing (China) [72] detected high concentrations of MP fibers (80% of the analyzed MP) in a size range smaller than 20 µm. In this study, samples were collected at three different height (0, 1.5, and 18 m) on the campus of the China University of Mining and Technology in Beijing, with active samplers on mixed cellulose ester filters (0.8 µm pore size). The collected samples were analyzed via SEM coupled with a dispersive X-ray detector. The reported MP fiber concentrations reached 5.7 × 10^−3^ particle mL^−1^. Air suspended MPs were also detected in remote areas up to 400 km offshore in the north-western Pacific Ocean [73]. Eleven atmospheric samples were collected with an active sampler device on glass microfiber filters (1.6 µm pore size) and then analyzed through µ-FTIR, attenuated total reflectance (ATR), and field emission SEM. MP abundance ranged from 0.0046 to 0.064 particles m^−3^ (on average 0.027 ± 0.018 particles m^−3^), with pelagic areas showing higher MP concentrations (0.037 ± 0.017 particles m^−3^) than nearshore locations (0.013 ± 0.007 particles m^−3)^. The detected polymers were predominantly fibrous rayon (67%) and PET (23%), followed by PE, PS, and polyvinyl chloride–polyvinyl alcohol (PVC-PVA) copolymers. Fibers made up most of the collected MPs (88–100%, average length = 853 µm), followed by fragments (0–8%), granules (0–6%), and films (0–2%). No correlation was found, in this study, between wind and MP distribution, while barometric pressure and relative humidity were negatively correlated with their abundance. Contrary to most studies, an investigation conducted in the metropolitan area of Hamburg (Germany) on atmospheric wet deposition [69] detected a higher abundance of fragments (95%, the majority < 63 µm) than fibers. Samples were collected with bulk samplers at six sites, biweekly, over a period of twelve weeks, and highlighted a median MP abundance of between 136.5 and 512.0 particle/(m^2^ day^−1)^ (mean of 275.0 particle/(m^2^ day^−1^)). Micro-Raman analyses showed that polyethylenes/ethylvinyl acetate copolymers were the most abundant MP polymers (48.8 and 22.0%, respectively). The rural sites in the southern area of Hamburg surprisingly showed the highest fragment concentration, and, overall, all sites showed a high time variation. Lastly, a study conducted by Sun and co-authors [67] in the area of Shanghai (China) aimed to quantify dry dispersion and wet deposition and correlate them with the values of PM_2.5_ and PM_10_. The sampling was conducted on eleven separate days with stainless steel buckets. The collected particles were first processed to remove organics, and then a density separation step was performed (ZnCl_2_); finally a subsample was analyzed via µ-Raman spectroscopy. MP abundances detected via wet deposition ranged from 1.1 × 10^3^ ± 0.06 × 10^3^ to 3.5 × 10^3^ ± 1.0 × 10^3^ particle/(m^2^ day^−1^) (mean of 2.1 × 10^3^ ± 1.0 × 10^3^ particle/(m^2^ day^−1^)). Of these, PE accounted for 49%, followed by PP (20%), PET (9.0%), and PA (7.1%). Similarly to what reported by Klein et al. [69], fragments accounted for the larger fraction of detected MPs, at approximately 72%, while fibers represented 28% of the wet sample. The concentrations detected via dry deposition were generally lower, except during the days with high PM values, and ranged from 0.91 × 10^3^ ± 0.09 × 10^3^ to 1.6 × 10^3^ ± 0.1 × 10^3^ particle/(m^2^ day^−1^) (mean of 1.2 × 10^3^ ± 0.2 × 10^3^ particle/(m^2^ day^−1^)). No difference was detected in the polymer composition compared to the samples acquired by wet deposition; however, a lower percentage of fibers (13%) was collected via dry deposition. These results confirm the impact of meteorological factors in influencing MP air concentration; in particular, the higher value of dry deposition in days with high PM_2.5_ and, to a lesser extent, PM_10_ concentration, highlighted the importance of wind action.

**Table 2 toxics-12-00320-t002:** Microplastics in outdoor environments.

Location	Mp Shape	Reported Concentration	References
Paris (France)	Fibers (80% between 100 and 500 µm)	Indoor environments (3–15 particle m^−3^) and outdoors (0.2–0.8 particle m^−3^).	Gasperi and co-authors 2018 [57]
Southern Iran	Fibers	From 0.3 to 1.1 particles m^−3^ and equally distributed between industrial and urban areas.	Abbasi et al., 2019 [62]
Ahvaz (Iran)	Fibers of size 15–35 µm	0.002–0.007 particle m^−3^, 23–341 particle g^−1^ of PM_10_ at site 1; 0.002–0.015 particle m^−3^, 34–162 particle g^−1^ of PM_10_ at site 2.	Abbasi et al., 2023 [71]
Shanghai (China)	Fibers	Average 1.42 ± 1.42 particles m^−3^ (annual weight of suspended MPs of 120.72 kg).	Liu et al., 2019 [63]
Beijing (China)	Fibers	5.7 × 10^−3^ particle mL^−1^	Li et al., 2020 [72]
North-western Pacific Ocean	Fibers (67%) Fragments (30%) granules (3%)	On average: 0.027 ± 0.018 particles m^−3^ Pelagic areas: 0.037 ± 0.017 particles m^−3^ Nearshore locations: 0.013 ± 0.007 particles m^−3^	Ding et al., 2022 [73]
Hamburg (Germany)	Fragments (95%, the majority < 63 µm)	136.5 and 512.0 particle/(m^2^ day^−1^) (mean of 275.0 particle/(m^2^ day^−1^))	Gaston et al., 2020 [64]
Shanghai (China)	Mainly fragments	Mean of 2.1 × 103 ± 1.0 × 10^3^ particle/(m^2^ day^−1^).	Sun et al., 2022 [67]

#### 5.4.2. Microplastic in Indoor Environments

Airborne contamination and the risks related to MP exposure for humans are particularly high and worrisome in indoor environments (Table 3). In fact, it has been estimated that people spend an average of 90% of their daily life in indoor environments, such as the home, offices, or on transportation [74]. Several sources of MPs in indoor air have been identified, including textiles, toys, rubber, kitchen items, electrical cables and electronics, and paint, as well as cleaning agents [75]. However, only a few studies have tried to estimate the number of MPs in indoor environments by focusing on two different matrices, i.e., settled dust or air.

Variable amounts of MPs with different shapes, sizes, and polymer compositions have been found in indoor settled dust. A pioneer study performed in three different indoor sites, including two private apartments and one office located at the University of Paris-Est-Créteil (Paris, France), has highlighted that the number of MPs, specifically fibers, in dust sampled from vacuum cleaner bags ranged between 1 and 60 fibers m^−3^ [76]. These amounts exceeded those measured in outdoor dust (range: 0.3–1.5 fibers m^−3^) [76]. Among these fibers, 33% were estimated to be made of plastic polymers, where polypropylene (PP) was predominant, falling within the category of MPs. The same study has also estimated the deposition rate of the fibers in indoor environments, which was calculated ranging between 1586 and 11,130 fibers/(day m^−2^), leading to an accumulation of fibers in settled dust ranging between 190–670 fibers mg^−1^ [23]. The investigation of indoor and outdoor MP contamination in dust samples from 39 Chinese cities has indicated the presence of polyethylene terephthalate (PET) MPs in all the analyzed samples (range 1550–120,000 µg g^−1^ indoors and 212–9020 µg g^−1^ outdoors), while polycarbonate (PC) MPs were detected in 70% of the samples, with median concentrations of 4.6 µg g^−1^ (indoors) and 2.0 µg g^−1^ (outdoors), respectively [63]. A further analysis of MP occurrence in house dust from 12 different countries worldwide has returned the presence of MPs in all the analyzed dust samples, with amounts ranging between 38 and 120,000 µg g^−1^ [77]. Nylon MPs, specifically polyamide 6 (PA6) and 66 (PA66), were detected in indoor dust from Chinese houses in the 0.431–86.3 µg g^−1^ and 3.10–92.9 µg g^−1^ ranges, respectively. Both PA6 and PA66 MP quantities measured in indoor dust exceeded those in the other environmental matrices, including sludge, marine, and freshwater sediments [78]. Lastly, Bahrina et al. [79] have investigated the relationship between the number of occupants and the number of MPs in indoor environments, i.e., in settled dust from offices, schools, and private apartments in Surabaya, Indonesia. The greatest numbers of MP were detected in offices (mean of 342 items on weekdays and 247 on weekends, respectively), while the lowest numbers were detected in apartments (mean of 120 items on weekdays and 111 on weekends, respectively). As expected, fibers in the 3000–3500 µm size range were the main source of MPs. The number of MPs collected during workdays exceeded that measured during the weekend, confirming the effect of activities and the number of occupants in indoor environments.

Some recent studies have investigated MP contamination in indoor air from different environments. The study performed by Vianello et al. [33] as investigated the exposure of humans to indoor airborne MPs through the application of a breathing thermal manikin. This device was placed in three private apartments, and all the collected samples were contaminated with MPs, whose concentrations ranged between 1.7 and 16.2 items m^−3^. The contamination fingerprint was characterized by fragments and fibers, accounting for, on average, 4% of the total identified items, while non-synthetic, natural items constituted the remaining items. Among synthetic polymers, polyester was the predominant one in all samples (81%), followed by polyethylene (5%) and nylon (3%).

Zhang et al. [80] investigated concentrations of MPs in indoor air from three different indoor microenvironments, such as a dormitory room, office, and a lecture building located at the East China Normal University. The highest MP abundance was detected in the dormitory (9.9 × 10^3^ MP/(m^2^ day^−1^)), followed by the office (1.8 × 10^3^ MP/(m^2^ day^−1^)), and the corridor (1.5 × 10^3^ MP/(m^2^ day^−1^)). Fibers represented the majority of MPs, whose polymer composition reflected that of the textile products used in the microenvironments. These results suggest that the amount and the fabric of textiles in the indoor microenvironment represent the main factors affecting MP abundance in the air.

An investigation of airborne MPs in indoor and outdoor air from coastal California has pointed out that MP concentrations in indoor air (3.3 ± 2.9 fibers and 12.6 ± 8.0 fragments m^−3^) were higher compared to outdoor air (0.6 ± 0.6 fibers and 5.6 ± 3.2 fragments m^−3^). In addition, indoor MP fragments (58.6 ± 55 µm) resulted as smaller than those collected outdoor (104.8 ± 64.9 µm), suggesting that the risk of exposure through inhalation in indoor environments is higher than outdoors because of both higher abundances and the smaller size of the MPs [64]. Lastly, a recent investigation of MP contamination performed in indoor air from 20 houses in Hull (UK) by Jenner et al. [81] showed that the mean MP quantity was 1414 MP/(m^2^ day^−1^), over a 6-month period. Fibers (5–250 µm size range) accounted for 90% of the MP pattern and were mainly composed of PET (62% of the identified items).

Overall, both in settled dust and air from indoor environments, the number of MPs exceeds that measured outdoors because of different sources of contamination and mechanisms involved in the dispersion of MPs, including speed ventilation, air flow, room partition, and climatic conditions [82]. All the studies confirmed the widespread dispersion of MPs indoors and suggest a potential risk to human health because of their daily inhalation or ingestions. This situation is particularly worrisome in working environments, especially for operators involved in the various steps of production, transformation, and use of polymers and plastic materials, mainly in the packaging, cosmetics, and textile sectors [83].

## 6. Potential Implications on Human Health

Atmospheric MPs could be compared to other particulate airborne contaminants, such as engineered nanomaterials (ENMs) or PM_2.5_ and PM_10_. ENMs are nanomaterials between 1 and 100 nm in size [84] that are commercially employed as nanotechnologies in several sectors, like in telecommunications, agrichemicals, and personal care products [85]. Given their widespread usage, these particles could become airborne and interact with several organisms and with humans. In terms of potential toxicity, given the large size difference and diverse chemical composition, atmospheric MPs and ENMs cannot be compared. Because of their smaller size, ENMs can enter organisms via inhalation, skin penetration, and ingestion, and affect them at the cellular level. Inhalation is the main pathway into the human body; once they enter the lungs, they can cross the blood–air-tissue barrier and enter the bloodstream, and so could potentially interact with several organs [85]. Particles belonging to the course PM (2.5–10 µm) group have a more similar size to MPs (which can also be included among the PM group); these particles can originate from vehicles, biomass burning, power plants, industrial emissions, and dust resuspension [86]. Exposure to PMs can have severe implications on human health [87]. They can lead to pulmonary and cardiovascular inflammatory response, impairment of the immune system in the lungs, and several respiratory infections; moreover, chronic PM_2.5_ exposure has been found to increase the risk of strokes and ischemic heart diseases [86,88].

### 6.1. Distribution of MPs in the Human Body

After examining the pathways through which MP enter the human body, it is necessary to analyze their fate within the body. MPs smaller than 10 μm in size can cross the cell membranes and enter the circulatory system, then spread to various tissues and organs. Recent studies have detected MPs in various body fluids, such as blood, saliva, and nasal secretions. In the placenta, MPs between 5 and 10 μm in size have been detected [89], as well as particles 50–500 μm in diameter [90]. In one study, the relationship between various parameters concerning the newborn (weight, height, and head circumference) and the amount of MPs in the placenta was also measured and calculated [91]. A significant and negative correlation was found between the abundance of MPs and these parameters, so it can be concluded that MPs can be transferred from the mother to the fetus, and they can cause toxic effects in the neonate [91].

The probability of fibrous MPs in the atmosphere entering our respiratory system varies according to their size. Their deposition in the respiratory tract decreases when the particle diameter reaches 5 mm [57]. The lungs are among the organs in which newly inhaled MPs can accumulate the most. The first evidence of their presence in lung tissue dates to 1998 [92]. Subsequent studies have found particles ranging in size from 1.60–5.56 mm of different compositions [63]. Some authors have investigated the presence of MPs in the upper respiratory tract of indoor and outdoor workers [93]. For this purpose, sputum and nasal lavage fluids were collected and analyzed, revealing that MPs can interact with the respiratory tract of both indoor and outdoor workers. The deposition of MPs in the lower respiratory tract was also examined. The first case study involved 18 nonsmokers aged 32 to 74 years from whom bronchoalveolar fluid samples were taken. Comparison with control samples consisting of an isotonic saline solution showed that the actual body fluid contained a far greater concentration of plastics [94]. The second case study investigated the relationship between smoking and inhalation of MPs by taking bronchoalveolar fluid samples from 17 smokers and 15 nonsmokers: the former group had a significantly higher concentration of MPs [95]. Both these studies constitute new evidence of the presence of MPs in the lower respiratory tract. MPs that enter the body through inhalation are undoubtedly more difficult to excrete than those introduced through the diet. Within the lungs, the surface area of the alveoli (about 150 m^2^) and the 1-milimeter-thick tissue allow particles to penetrate the cardiovascular system and disperse within the body.

Another recent study has confirmed the presence of MPs within human blood and provides clear evidence that they can migrate into the body via the bloodstream [96]. The discovery of MPs inside blood clots indicates that the impact of exposure on human health should by no means be underestimated. The accumulation of exogenous particles, including pigment microparticles and MPs in thrombi, has been validated by Raman spectra. The results of this study indicated that the effects of exogenous factors on thrombosis could not be ignored [97]. The first study with the aim of investigating the presence of MPs in feces was in 2018 [98], showing that, in humans, they are ingested unknowingly, then reach the intestines and are partially eliminated through the feces. Samples from eight healthy volunteers aged 22–65 years were analyzed; all of them tested positive for the presence of MPs, specifically nine types of plastics, mostly polypropylene and polyethylene terephthalate. In addition, an average of 20 MPs per 10 g of feces was found [99]. A recent study that analyzed meconium and feces samples has proved that infants are exposed to higher levels of MPs than adults [100].

So far, MPs have been found in human feces and in the colon, lungs, placenta, breast milk, blood, liver, spleen, kidneys, and skin on the hands and face. This is possible because, after being taken orally, MPs can be absorbed in the gastrointestinal tract and reach other tissues and organs. Of these, polypropylene, polyethylene, polyvinyl chloride, and polystyrene are the most abundant, being those to which humans are most frequently exposed in daily life [41].

A current study [101] has shown that 20–100 mm MPs can concentrate in all human tissues, with polyvinyl chloride (PVC) being the most abundant polymer. The highest concentration of MPs was detected in lung tissue, followed by that in the small intestine, large intestine, and tonsils.

### 6.2. Toxicity

The intake of MPs into the body potentially exposes humans to many risks, including the possibility of damaging various barriers, inducing oxidative stress, regulating gene expression, impairing the functions of certain organs, and developing cancer [43]. MPs inhaled and accumulated in the lungs can lead to long-term damage to the organ and the membrane that protects it, alteration in its morphology, an inflammatory response, and dysfunction [55].

One of the key factors influencing microplastics’ toxicological effects is the particle size, in addition to the type, shape, and concentration. In general, the smaller the particle size, the more toxic they are to organisms [102]. Small MPs can penetrate the blood–brain barrier, leading to increased levels of ROS (reactive oxygen species) and MDA (malondialdehyde) and a significant decrease in glutathione (GSH) levels [103]. Therefore, MPs can induce oxidative stress within nerve tissue [103] in mice. Thyey can also reduce the expression of connectin, a protein present in the blood–brain barrier, stimulating the production of reactive oxygen species that induce nerve cell apoptosis and micro-thrombosis phenomena, leading to neurological dysfunction [104] (Figure 5).

MPs can also exert toxicity at the liver and metabolic levels [104]. When MPs reach the liver through the bloodstream, they alter the normal functioning of the organ and induce DNA damage and release in the nucleus and mitochondria of liver cells, activating an inflammatory response, such as the expression of pro-inflammatory cytokines and leading to liver fibrosis [105]. Moreover, aggregation of MPs at the level of liver tissue inhibits the accumulation of fatty acids and their methyl and ethyl esters. This condition destroys regular lipid metabolism, causing hepatic steatosis [106]. Disorders in amino acid and glucose metabolism were also observed [107]. At the same time, MPs may have adsorbed toxic substances, such as cadmium, inducing organ death due to heavy metal poisoning [108].

Ingestion of high concentrations of MPs can disrupt the balance of the gut flora, altering the abundance and diversity of microorganisms [72]. This can cause the release of some toxic products of bacterial metabolism, leading to an inflammatory condition. The kidney could be one of the organs impacted by MP aggregation. Its exposure can cause significant damage, such as oxidative stress-induced effects, leading to an inflammatory response and tissue injury [109].

In addition, MPs have been reported to reduce reproductive capacity in both male [110] and female [111] rat specimens. Because of their small size, MPs possess a high surface area/volume ratio. Materials with a large surface area are highly cytotoxic to cells and tissues and can damage DNA within the cell nucleus. This damage leads to DNA mutations that cause cancer [112]. Carcinogenic effects are included in genotoxic effects along with teratogenic and mutagenic ones. Furthermore, MPs can adsorb hydrophobic organic contaminants, giving rise to so-called “Trojan Horse mechanism” effects [113], which are themselves carcinogenic [114]. Heavy metals used in plastic production, such as arsenic, cadmium, chromium, mercury, and lead, are also carcinogens according to the International Agency for Research on Cancer (IARC).

## 7. Conclusions and Future Perspectives

Considering the dearth of information regarding both human exposure and the toxicological risks related to MPs, an accurate risk assessment is currently not possible. An estimation of exposure to MPs is essential, but there are still no sensitive and validated methods to detect trace amounts of very small items. The collection of further new data of interest is, therefore, significant for studying the possible long-term exposure of the general population to low concentrations of MPs, and for the development and standardization of new methods for assessing the state of pollution from MPs and/or chemical additives present in different environmental compartments.

This review has evaluated the state-of-the-art research on the critical issue of MPs, highlighting the living and working environments where the high exposure of workers to dust, vapors, and dangerous gases can occur [107]. It is of particularly important to investigate MP exposure, particularly in the workplace, where regulations for nano- and microplastic workplace contaminants are not yet present. Our ambitious goal will be the development of operative protocols for minimizing the risk from occupational exposure to MPs. In the meantime, it is important to raise awareness among workers of MP pollution, promoting a sustainable development, as adopted by the European strategy, which aims to manage and reduce plastic waste.

## Figures and Tables

**Figure 1 toxics-12-00320-f001:**
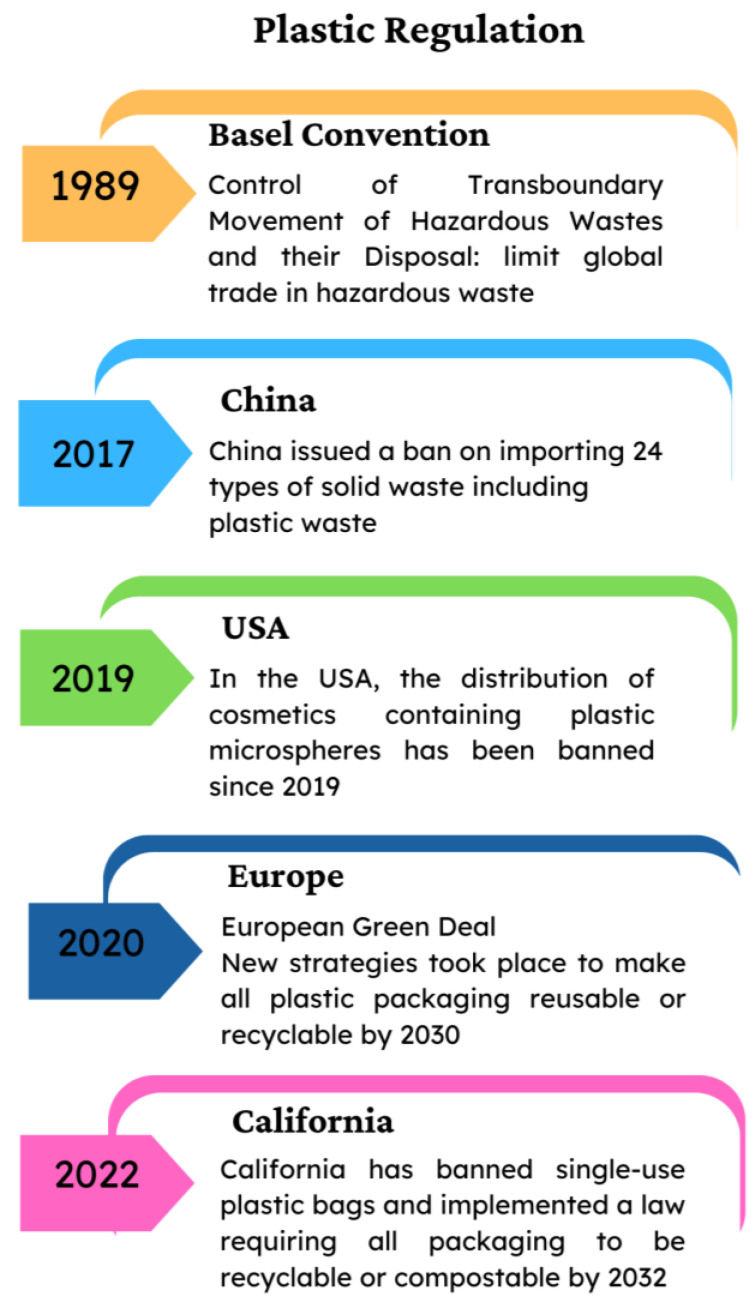
Regulation of plastic pollution in different regions.

**Figure 2 toxics-12-00320-f002:**
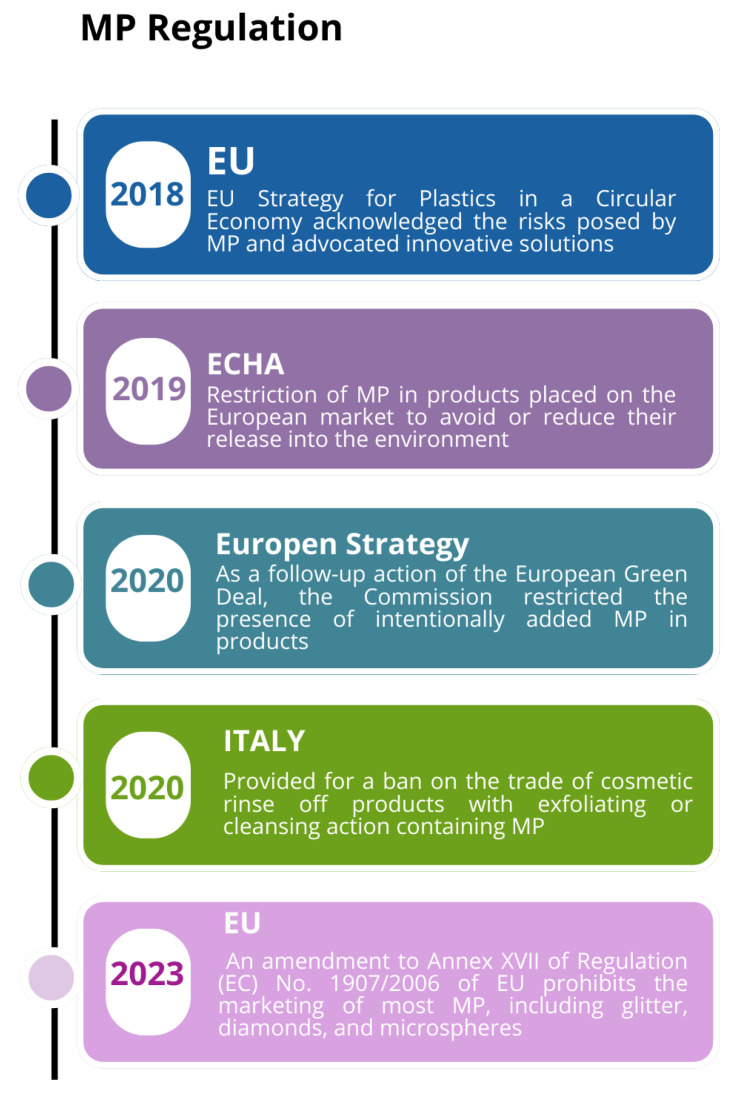
Main MP legislation in different regions.

**Figure 4 toxics-12-00320-f004:**
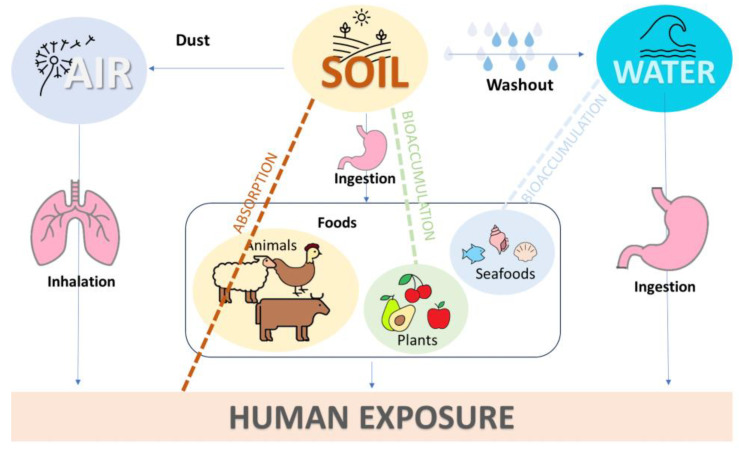
The presence of microplastics in the environment directly results in human exposure through several routes, such as diet, drinking water, and air. The arrows represent the exposure routes and the dotted lines the accumulation process that impact human health.

**Figure 5 toxics-12-00320-f005:**
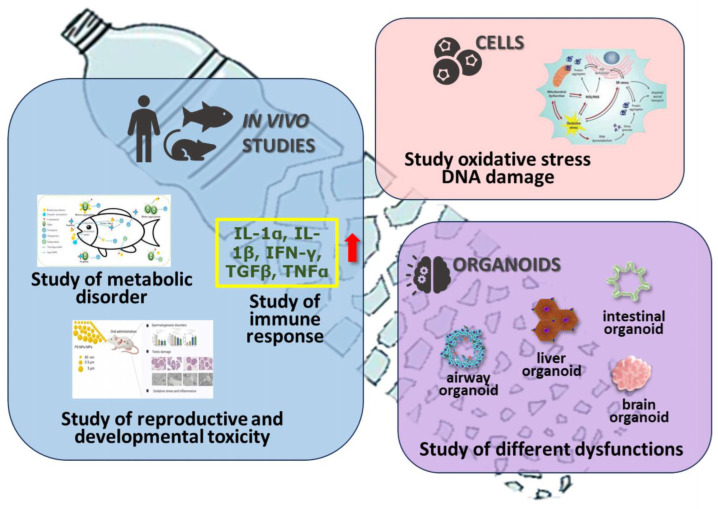
The potential effects of microplastics for animal and human health: the possibility of damaging various barriers, inducing oxidative stress, regulating gene expression, impairing the functions of certain organs, and developing cancer and neurological dysfunction. The red arrow around the yellow frame represents the impairment of the immune system in the lungs: inhalation of MP may cause the upregulated expression of the inflammatory protein (TGF-β and TNF-α, IL-1β, IL-1α and IFN-γ) in lung tissue of rats and mice.

**Table 1 toxics-12-00320-t001:** Analytical methods used to determine MPs in environmental matrices.

Analytical Methods	Advantages	Disadvantages
**Stereomicroscopy**	First visual analysis of potential MPs (shape and size) Improving MP detection using fluorescent dyes (Nile Red, Rose Bengal)	No information about the chemical composition of particles Possibility of confusing sand or carbon particles as MPs or vice versa For items smaller than 10 μm, difficulties discriminating if they have polymeric or mineral compositions
**SEM**	More detailed information about the morphological characteristics of samples	No information about the chemical composition of particles
**SEM-EDS**	High-resolution images Determination of the elemental composition of particles Discrimination of particles of inorganic origin from organic ones	In general, visual identification is now considered outdated because of insufficient results and false positives (valid also for stereomicroscopy and SEM analysis)
**Pyr-GC/MS**	Simultaneously analyzing all MPs by quantifying their total mass Information about the chemical composition of samples	No information about the shape, size, or number of items Destructive method
**Vibrational** **FTIR-Spectroscopies**	Immediate identification of functional groups present in a molecule by observing the spectrum Identify functional groups with polarized bonds, e.g., C=O Possibility to be combined with a microscope for more specific analysis Spectra comparison with databases to identify molecules Simple data recording techniques	Susceptible to the presence of additives on the polymer surface Dust, microbial fouling, and dyes can interfere with the signals Water interferes with the sample analysis Smaller particles are more complex to analyze and require FTIR microspectroscopy
**Raman Spectroscopies**	Possibility to be combined with a microscope for more specific analysis Raman microscopy is an indispensable tool for the analysis of very small microplastics Determine vibrational modes of molecules and provide a structural fingerprint by which molecules can be identified More sensitive to vibrations of homopolar bonds, such as C-C Water does not interfere with the analysis: signals are narrower and easier to identify Simple data recording techniques	Susceptible to the presence of additives on the polymer surface Dust, microbial fouling, and dyes can interfere with the signals

**Table 3 toxics-12-00320-t003:** Microplastic in indoor environments.

Location	MP Shape	Reported Concentration	References
University of Paris-Est-Créteil (Paris, France)	Fibers	- From 1 to 60 fibers m^−3^ - From 1586 to 11,130 fibers/(day m^−2^), accumulation in settled dust between 190–670 fibers mg^−1^	Dris et al., 2018 [76]
39 Chinese cities	Fibers	PET: range 1550–120,000 µg g^−1^ (indoors); 212–9020 µg g^−1^ (outdoors); PC: median concentrations 4.6 µg g^−1^ (indoors) and 2.0 µg g^−1^ (outdoors)	Liu et al., 2019 [63]
12 different countries worldwide	Fibers ?	Ranging between 38 and 120,000 µg g^−1^	Zhang et al., 2020 [77]
Chinese houses	Nylon fibers	0.431–86.3 µg g^−1^ and 3.10–92.9 µg g^−1^ ranges	Peng et al., 2020 [78]
Surabaya (Indonesia)	Fibers	Mean of 342 items on weekdays and 247 on weekends, respectively, in apartments (mean of 120 items on weekdays and 111 on weekends)	Bahrina et al., 2020 [79]
Aarhus (Denmark)	Fibers, fragments	Between 1.7 and 16.2 items m^−3^	Vianello et al., 2019 [33]
East China Normal University	Fibers	Dormitory (9.9 × 10^3^ MP/(m^2^ day^−1^)), followed by the office (1.8 × 10^3^ MP/(m^2^ day^−1^)), and the corridor (1.5 × 10^3^ MP/(m^2^ day^−1^))	Zhang et al., 2020 [80]
Coastal California	Fibers	In indoor air (3.3 ± 2.9 fibers and 12.6 ± 8.0 fragments m^−3^), values were higher compared to outdoor air (0.6 ± 0.6 fibers and 5.6 ± 3.2 fragments m^−3^)	Gaston et al., 2020 [64]
Hull (UK)	Fibers	Mean amount was 1414 MP/(m^2^ day^−1^)	Jenner et al., 2021 [81]

## Data Availability

No new data were created or analyzed in this study. Data sharing is not applicable to this article.

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
