# Peer review of "Potential Effects of Environmental and Occupational Exposure to Microplastics: An Overview of Air Contamination"

_toxics, 2024, doi:10.3390/toxics12050320_

Round 1
Reviewer 1 Report
Comments and Suggestions for Authors
This review highlights the presence of microplastic (MP) in the environment and the subsequent human exposure, which has prompted increased attention and research in recent years. It aims to summarize the current state of legislation, contamination in indoor and outdoor environments, and potential health risks for humans due to exposure to airborne MP. The topic is important; however, there are numerous questions in this manuscript that need major revisions before acceptance. The specific comments are as follows:
Abstract: the background introduction (Line 12-23) is too long. More solid conclusion of your review should be added here, as well as the sigificance of this review.
Line 38: the reference of "European Parliament" should be given here. Maybe a website.
Line 64: this section "Regulation on microplastic pollution" is important and attractive. A figure of timeline of the regulations in different countries should be added here to clearly illustrate the regulations. Besides, please pay attention if the regulation aims to control plastic pollution or microplastic pollution.
Line 190-191: this figure 1 is too simple to get the information. Maybe you can list the detection limits of MP in these different matrices directly in this figure.
Line 603: firstly, where do the figures come from? please give the reference here. In addtion, the examples of metabolic disorder and reproductive and developmental toxicity are driven from fish or rat, which are different from humans. Please change the exmaples and revise the corresponding sentences in the manuscript.
Comments on the Quality of English Language
The english used is correct and readable.
Author Response
Please, see the attached file for the comments to the revision process.

Reviewer 2 Report
Comments and Suggestions for Authors
The review is timely and discusses an important topic in the field of toxicology. Overall appearance, however, must be improved to make this review better organized and be attractive to readers. More details on this and other recommendations are below.
1. Discussion where the comparison of microplastic and other particulate pollutants such as PM2.5 and engineered nanomaterials (ENM) will make this review more comprehensive and would put it in the context of prior knowledge.
2. Nowadays, there is much discussion about the size change of microplastics and appearance of the nanoplastics. As the size effect is very important from the toxicological viewpoint, an extensive discussion is recommended.
3. Information in many chapters should be better presented not as bare text, but structured by summarizing into tables highlighting important aspects. For instance, there might be a comparison table about the regulation of plastic pollution in different regions (Figure 2), a comparison table of different analytical methods with advantages and drawbacks (Chapter 4), etc. The exposure levels of different microplastics (Section 5.4) can be summarized into Tables or even Figures to be more illustrative.
4. Concept Figures 1 and 2 must contain a drawing of nanoplastics for better understanding. Figure 1 appears to be too general and authors are encouraged to consider improvement to show more aspects of it.
5. The part in lines 247-250 on page 6 seems to be a remained part from the article template with some guidance for authors and must be removed.
6. The toxicity of microplastics (Section 6.2) should be discussed in comparison to other known toxic micro- and nanoparticles such as engineered nanomaterials.
7. The conclusions are not of general interest and must be reconsidered and revised to explain in particular future perspectives of microplastic exposure increase and possible health effects depending on regulations. On the other hand, the explanation about the INAIL research project sounds like a self-promotion of the authors and must be avoided in the conclusions.
Author Response

(The authors gave the same response as above.)

Round 2
Reviewer 1 Report
Comments and Suggestions for Authors
The authors carefully revised their paper, greatly improving its quality.
Author Response
We changed the dimension of Fig 1 and 2. We corrected the text in Fig 3
Reviewer 2 Report
Comments and Suggestions for Authors
The manuscript has been significantly improved and looks much attractive. The two minor points are as follows.
1. The size of Figures 1 and 2 can be made smaller
2. There is a mistake in Figure 3, APPROCH → APPROACH
Author Response

(The authors gave the same response as above.)
